# IA-Body Composition CT at T12 in Idiopathic Pulmonary Fibrosis: Diagnosing Sarcopenia and Correlating with Other Morphofunctional Assessment Techniques

**DOI:** 10.3390/nu16172885

**Published:** 2024-08-28

**Authors:** Rocío Fernández-Jiménez, Alicia Sanmartín-Sánchez, Eva Cabrera-César, Francisco Espíldora-Hernández, Isabel Vegas-Aguilar, María del Mar Amaya-Campos, Fiorella Ximena Palmas-Candia, María Claro-Brandner, Josefina Olivares-Alcolea, Víctor José Simón-Frapolli, Isabel Cornejo-Pareja, Patricia Guirado-Peláez, Álvaro Vidal-Suárez, Ana Sánchez-García, Mora Murri, Lourdes Garrido-Sánchez, Francisco J. Tinahones, Jose Luis Velasco-Garrido, Jose Manuel García-Almeida

**Affiliations:** 1Department of Endocrinology and Nutrition, Virgen de la Victoria University Hospital, 29010 Malaga, Spain; rociofernandeznutricion@gmail.com (R.F.-J.); isabel.mva13@gmail.com (I.V.-A.); mariadelmarac2@gmail.com (M.d.M.A.-C.); victorsimonfrapolli.med@gmail.com (V.J.S.-F.); isabelmaria_cornejo@hotmail.com (I.C.-P.); pguirado1991@gmail.com (P.G.-P.); alvarovidal1992@gmail.com (Á.V.-S.); anaasanchez12@gmail.com (A.S.-G.); moramurri@gmail.com (M.M.); lourdes.garrido@ibima.eu (L.G.-S.); fjtinahones@uma.es (F.J.T.); jgarciaalmeida@gmail.com (J.M.G.-A.); 2Instituto de Investigación Biomédica de Málaga y Plataforma en Nanomedicina-IBIMA Plataforma BIONAND, 29010 Malaga, Spain; 3Department of Medicine and Dermatology, Málaga University, 29016 Malaga, Spain; 4Department of Endocrinology and Nutrition, Quironsalud Málaga Hospital, Av. Imperio Argentina, 29004 Malaga, Spain; 5Department of Endocrinology and Nutrition, Son Espases University Hospital, 07120 Mallorca, Spain; jolivares@ssib.es; 6Department of Neumology, Virgen de la Victoria University Hospital, 29010 Malaga, Spain; evacabreracesar@gmail.com (E.C.-C.); jlvelascogarrido@hotmail.com (J.L.V.-G.); 7Department of Neumology, Regional University Hospital, 29010 Malaga, Spain; fespildorahernandez@gmail.com; 8Endocrinology and Nutrition Department, Vall D’Hebron University Hospital, 08035 Barcelona, Spain; fiorellaximena.palmas@vallhebron.cat; 9Endocrinology and Nutrition Department, Clinic Hospital, 08036 Barcelona, Spain; mclaro95@gmail.com; 10Centro de Investigación Biomédica en Red Fisiopatología de la Obesidad y Nutrición (CIBEROBN), Carlos III Health Institute (ISCIII), University of Málaga, 29010 Malaga, Spain; 11Instituto de Investigación Biomédica de Málaga y Plataforma en Nanomedicina-IBIMA Plataforma BIONAND, Heart Area, Victoria Virgen University Hospital, 29010 Malaga, Spain

**Keywords:** idiopathic pulmonary fibrosis, morphofunctional assessment, body composition, sarcopenia, malnutrition, bioelectrical impedance analysis, nutritional ultrasound, computed tomography, mortality

## Abstract

Background: Body composition (BC) techniques, including bioelectrical impedance analysis (BIVA), nutritional ultrasound^®^ (NU), and computed tomography (CT), can detect nutritional diagnoses such as sarcopenia (Sc). Sc in idiopathic pulmonary fibrosis (IPF) is associated with greater severity and lower survival. Our aim was to explore the correlation of BIVA, NU and functional parameters with BC at T12 level CT scans in patients with IPF but also its relationship with degree of Sc, malnutrition and mortality. Methods: This bicentric cross-sectional study included 60 IPF patients (85.2% male, 70.9 ± 7.8 years). Morphofunctional assessment (MFA) techniques included BIVA, NU, CT at T12 level (T12-CT), handgrip strength, and timed up and go. CT data were obtained using FocusedON^®^. Statistical analysis was conducted using JAMOVI version 2.3.22 to determine the cutoff points for Sc in T12-CT and to analyze correlations with other MFA techniques. Results: the cutoff for muscle area in T12-CT was ≤77.44 cm^2^ (area under the curve (AUC) = 0.734, sensitivity = 41.7%, specificity = 100%). The skeletal muscle index (SMI_T12CT) cutoff was ≤24.5 cm^2^/m^2^ (AUC = 0.689, sensitivity = 66.7%, specificity = 66.7%). Low SMI_T12CT exhibited significantly reduced median survival and higher risk of mortality compared to those with normal muscle mass (SMI cut off ≥ 28.8 cm/m^2^). SMI_T12CT was highly correlated with body cell mass from BIVA (r = 0.681) and rectus femoris cross-sectional area (RF-CSA) from NU (r = 0.599). Cronbach’s α for muscle parameters across different MFA techniques and CT was 0.735, confirming their validity for evaluating muscle composition. Conclusions: T12-CT scan is a reliable technique for measuring low muscle mass in patients with IPF, specifically when the L3 vertebrae are not captured. An SMI value of <28.8 is a good predictor of low lean mass and 12-month mortality in IPF patients.

## 1. Introduction

Idiopathic pulmonary fibrosis (IPF), a chronic lung disease with progressive and irreversible interstitial fibrosis [1], has been associated with sarcopenia rates of 22.9–39.3% [2,3] and malnutrition rates of 18.5% to 55% [4,5,6]. Patients with chronic respiratory diseases (COPD) are prone to malnutrition due to increased energy requirements from heightened respiratory muscle activity, inflammation mediators or hypoxemia [7,8]. Additionally, in IPF, given its complex and multifactorial nature, prognosis may be further exacerbated by several comorbidities [9], which can contribute to malnutrition. Despite the fact that new antifibrotic medications (pirfenidone and nintedanib) have been developed and appear to slow disease progression, mortality rate remains devastating [1,10,11]. Çinkooğlu et al. reported from a cohort of 195 patients a second-year mortality of 33.8% and a mean survival time post-diagnosis of 66.04 ± 6.47 months [11].

Sarcopenia (Sc), as defined by the EWGSOP2 criteria [12,13], is a nutritional diagnosis characterized by low muscle strength in the hand grip strength test (HGS), low muscle quantity or quality measured in bioelectrical impedance vector analysis (BIVA) or computed tomography (CT), and often low physical performance, which is evaluated with a timed up and go test (TUG) and the six-minute walk test (6 MWT). Patients with Sc have been reported to have a higher incidence of disability, increased risk of falls, poor quality of life, increased intensive care unit length of stay [14], higher hospital complications [15] and higher mortality rates [12,15,16,17]. Specifically, in IPF, whose prevalence of Sc at diagnosis was 23% [2], Sc has been associated with a significantly higher severity of the disease [2] and lower median survival [11,18].

Highly related to Sc, malnutrition in IPF patients is linked to higher mortality, increased hospitalizations, and lower quality of life [5,6,19,20], and it has even been reported as a prognostic indicator of antifibrotic therapy discontinuation [20]. Currently, the diagnosis of malnutrition is made according to GLIM criteria, requiring one phenotypic criterion (weight loss, low body mass index (BMI) or reduced muscle mass) and one etiologic criterion (reduced food intake/assimilation or the presence of inflammation) [21]. Often, BMI or weight loss are used as phenotypic criteria instead of a thorough diagnostic evaluation that assesses muscle strength, muscle quantity, and physical performance, which may offer greater utility than multiple tests in promptly identifying and addressing early nutritional impairments [2,8,22,23,24,25,26].

Consequently, a comprehensive morphofunctional assessment (MFA) [25,26] is crucial for ensuring proper and individualized nutritional care [8]. To evaluate muscle quantity, the most available techniques are nutritional ultrasound^®^ (NU) [27], BIVA and CT. Phase angle (PhA) by BIVA [22] and rectus femoris cross-sectional area (RF-CSA) by NU [27] are shown to be good prognostic markers of 12-month mortality [28].

However, evidence on the feasibility of MFA with CT is limited [29]. CT scans at the L3 level (L3-CT) have been considered the best single-slice CT for estimating whole-BC [29,30] and show a high correlation with BIVA and dual-energy X-ray absorptiometry [31]. They are used to diagnose Sc in various pathologies [14,23,24,32,33,34]. In IPF patients, a study identified a strong association between the cross-sectional area of the erector spinae muscle area (ESMCSA) at L3-CT and clinical parameters such as forced vital capacity (FVC) and forced expiratory volume in the first second (FEV1) [11].

Nonetheless, L3-CT is available in abdominal CT but not in thoracic CT (which is often used for monitoring IPF patients), which is why some studies have compared L3-CT and CT scans at the T12 level (T12-CT), as it is the closest vertebra to L3 and is always captured in thoracic CT, in searching for the best thoracic vertebra for BC [14,30,33,35,36,37]. In IPF patients, low ESMCSA at T12-CT was found to be a significant predictor of all-cause mortality [38], highlighting its importance as a key prognostic indicator, surpassing others such as BMI and FVC [39].

Our aim is to explore whether the BC assessed by T12-CT correlates with other morphofunctional techniques such as BIVA and NU, as well as functional tools like TUG, 6 MWT and respiratory parameters. Additionally, we would like to assess its relationship with outcomes such as mortality, degree of Sc and malnutrition.

## 2. Materials and Methods

### 2.1. Setting Study

A prospective, observational, bicenter study of routine clinical practice was conducted in the nutrition unit of the Endocrinology and Nutrition Unit of the Virgen de la Victoria University Hospital. Our sample was constituted of patients with different stages of idiopathic pulmonary fibrosis diagnosed, thanks to the biannual assessment usually performed in the pneumology service of Virgen de la Victoria University Hospital. Patients were also received from the Regional University Hospital of Málaga. The sample was followed for twelve months, from April 2021 to July 2022.

All subjects gave their informed consent for inclusion before they participated in the study. The study adhered to the Declaration of Helsinki and received approval from the Ethics Committee of Málaga on 5 April 2022 (reference number 1743-N-21).

Every patient enrolled in our study met the inclusion criteria (IPF diagnosticated, consent to participate in the study via accepted informed consent and a CT scan performed within 3 months before or after the initial nutritional assessment), and none of the exclusion criteria (refusal to participate or the inability to perform BIVA measurements for reasons related to ethnicity, extensive skin lesions, the extravasation of fluids through the route and local hematomas, amputation, etc., or a life expectancy of fewer than 3 months). A flow chart detailing patient selection is shown in Appendix A.

Clinical data including comorbidities, treatments, smoking, mortality and respiratory parameters such as forced vital capacity (FVC) and carbon monoxide diffusing lung capacity (DLCO) were collected from clinical records [40]. Patients were classified after a 12-month follow-up (between April 2021 to July 2022) according to GLIM criteria: non malnutrition, moderate malnutrition and severe malnutrition.

### 2.2. Anthropometric and Body Composition Measurements

#### 2.2.1. Bioelectrical Impedance Vector Analysis

BC was assessed using a 50 kHz phase-sensitive impedance analyzer (BIVA 101 Whole Body Bioimpedance Vector Analyzer, AKERN, Florence, Italy) delivering 800 μA [41,42] via tetrapolar electrodes positioned on the right hand and foot. Body weight and height were measured, and BIVA measurements were taken with patients in a supine position after a 5 min rest [42].

Phase angle (PhA) was calculated and standardized against age and sex-matched references using data from healthy Italian adults [41,43].

The BIVA device’s technical accuracy was evaluated daily using a precision track, where all measured values of R (resistance) and Xc (reactance) were consistently ±1 Ω of the reference value of 385 Ohm. In vivo reproducibility showed a 1–2% coefficient of variation (CV) for R and Xc [44].

#### 2.2.2. Nutritional Ultrasound^®^ [27]

Rectus femoris quadriceps muscle ultrasonography (RF-CSA) of the lower extremity was performed, in a supine position, with a 10–12 MHz probe and a multifrequency linear array (Mindray Z60, Madrid, Spain) by a trained clinician. We measured, in a supine position, the anteroposterior muscle thickness at the level of the lower third from the superior pole of the kneecap and the anterosuperior iliac spinous. Parameters collected were rectus femoris axis (RF-*Y*-axis and RF-*X*-axis), circumference (RF-CIR), cross-sectional area (RF-CSA), and subcutaneous fat of the leg (L-SAT), each one was made three times to use for the analysis the mean.

For abdominal adipose tissue assessment, we measured at the midpoint between the xiphoid process and the navel to obtain T-SAT (total subcutaneous abdominal fat), S-SAT (superficial adipose subcutaneous abdominal fat) and VAT (preperitoneal or visceral fat) in centimeters.

#### 2.2.3. Functional Assessment

HGS was tested with a JAMAR hand dynamometer (Asimow Engineering Co., Los Angeles, CA, USA) in a seated position with the elbow of the dominating arm flexed to 90 degrees. Patients performed three maximal isometric contractions, and we registered the median value [45,46].

TUG is performed by measuring the time, in seconds, to stand up from a wheelchair, walk 3 m, turn round to the chair and sit again and 6 MWT consists of how many meters the patient traveled in 6 min. Both tests were selected to evaluate the functional assessment for its extensive evidence [47,48].

#### 2.2.4. Computed Tomography at T12 Level by FocusedON^®^ [49,50]

Regarding T12-CT, axial sections at that vertebra level of CT images were provided from available CT imaging performed in clinical practice within three months of the initial nutritional assessment. The muscles included in the analyses were psoas, erector spinae, quadratus lumborum, transversus abdominis, external and internal obliques and rectus abdominis. Adipose tissue was assessed and classified as subcutaneous, visceral and intramuscular.

We assessed skeletal muscle and abdominal adipose tissue area, including the T12 skeletal muscle area (SMA_T12CT, cm^2^ and %), the T12 muscle index (SMI_T12CT, cm^2^/m^2^, obtained by adjusting the measured muscle areas to patients’ heights), the intramuscular adipose tissue area (IMAT, cm^2^ and %), the subcutaneous fat area (SFA) (cm^2^ and %), and the visceral fat area (VFA) (cm^2^ and %). The mean attenuation of each segmented tissue was measured in Hounsfield units (HU). Skeletal muscles were constituted by psoas, quadratus lumborum, erector spinae, external and internal obliques, transversus abdominis and rectus abdominis.

The CT images centered at the T12 vertebrae level were analyzed using FocusedON software (https://focusedon.es, accessed on 25 August 2024). from ARTIS Development, a medical image segmentation tool based on artificial intelligence. FocusedON enables automatic tissue segmentation, with the option for manual corrections as needed. Typical Hounsfield Unit (HU) thresholds used in these corrections are: −29 to +150 for skeletal muscle mass (SMM), −190 to −30 for subcutaneous adipose tissue (SAT) and intermuscular adipose tissue (IMAT), and −150 to −50 for visceral adipose tissue (VAT). Cross-sectional areas (cm^2^) for each tissue type are automatically calculated based on the pixel size and the number of labeled pixels for each tissue. The Hounsfield Unit (HU) value provided for each tissue corresponds to the average HU values of the labeled pixels for that tissue.

It is worth mentioning that the T12 vertebrae level was chosen as it is the closest to the L3, which is consistently captured in most thoracic CT scans, unlike L1 or L2, which may not be present in many cases. Additionally, at this level, there are no major vessels, such as the thoracic aorta, which contain muscle in their structures and could potentially interfere with our software’s analysis, unlike the T4 level, which has also been examined in other studies.

#### 2.2.5. Assessment of Malnutrition and Sarcopenia

Diagnosis of malnutrition was performed using the Global Leadership Initiative on Malnutrition (GLIM) criteria. All patients were found to meet at least one etiological criterion, as they had IPF, which is considered a chronic inflammatory condition. For the phenotypic criteria, patients were identified based on low BMI, weight loss greater than 5% over six months, and low Fat-Free Mass Index (FFMI) as measured by BIA, according to the cutoff points recommended by the consensus [21].

To diagnose Sc, we used EWGSOP2 criteria [12,13]. HGS < 27 kg for men and <16 kg for women indicated probable Sc. If it was combined with low muscle mass, measured by BIVA, with ASMM (appendicular skeletal muscle mass by BIVA) < 20 kg for men and <15 kg for women or ASMI (appendicular skeletal muscle mass index) < 7.0 kg/m^2^ for men or <5.5 kg/m^2^ for women, the diagnosis of Sc was made.

#### 2.2.6. Statistical Analysis

Statistical analysis was conducted using JAMOVI (version 2.3.28 for macOS). Descriptive variables included normally distributed continuous variables (presented as means and standard deviations) and categorical variables (presented as percentages). The evaluation of the predictive property of muscle mass variables was based on the receiver operating characteristic curve (ROC) and AUC (area under the curve). The diagnostic accuracy of SMA_T12CT and SMI_T12CT to identify optimal cut-off values for Sc and low muscle mass was assessed using receiver operating characteristic (ROC) curves. Additionally, the area under the curve (AUC) was calculated to determine the test’s discriminative power, with AUC accuracy estimated by plotting sensitivity versus specificity. Correlation between all morphofunctional techniques was evaluated using the Pearson correlation coefficient, with parameters categorized into muscle and adipose tissue parameters. Scale reliability was assessed using Cronbach’s alpha. To determine the probability of survival and the risk of death in patients with low muscle mass, we utilized the Kaplan–Meier product-limit estimate over a twelve-month period. Kaplan–Meier survival curves were compared using the log-rank (Mantel-Cox) test. To assess the association between SMA_T12CT and SMI_T12CT with mortality in IPF patients, Cox proportional hazards regression was performed, calculating hazard ratios and their confidence intervals. Statistical significance was considered at *p* < 0.05.

## 3. Results

The study evaluated various demographic, anthropometric, respiratory, functional and BC parameters in 60 subjects in the subgroup who were predominantly male (85.2%) with a mean age of 70.9 ± 7.8 years and a mean BMI of 27.7 ± 3.7 kg/m^2^. A comparison was made between sarcopenic (*n* = 12) and non-sarcopenic (*n* = 48) individuals (Table 1). Overall, there were significant differences between the non-sarcopenic and sarcopenic groups. The sarcopenic group was older, had a higher percentage of males and showed significant differences in HGS, SMI_T12CT, BCM, RF-CSA, and RF-*Y*-axis with *p*-values all less than 0.05. In order to assess the effect of the sample, a hedge analysis was carried out, as shown in Figure 1.

### 3.1. Sarcopenia Criteria (EWGSOP2) 

Table 2 presents the EWGSOP2 criteria for Sc by sex, evaluating HGS, ASMM and ASMI among 60 subjects. Low HGS was observed in 19 subjects (31.7%, *p* < 0.001), low ASMM was identified in 30 subjects (49.2, *p* = 0.009) and low ASMI was found in 28 subjects (45.9%, *p* = 0.139). 

### 3.2. Body Composition Parameters by T12-CT by Sarcopenia Criteria

In Table 3, significant differences were observed in several BC parameters by T12-CT between sarcopenic and non-sarcopenic individuals. Specifically, sarcopenic individuals had lower SMA_T12CT (60.6 ± 12.9 cm^2^ vs. 78.8 ± 22.3 cm^2^, *p* = 0.009), SMI_T12CT (22.6 ± 4.8 cm^2^/m^2^ vs. 27.2 ± 7.1 cm^2^/m^2^, *p* = 0.035), and VAT area (123.5 ± 36.3 cm^2^ vs. 191.5 ± 84.8 cm^2^, *p* = 0.009), while the SAT area was higher in sarcopenic individuals (152.4 ± 77.3 cm^2^ vs. 111.6 ± 48.3 cm^2^, *p* = 0.025).

### 3.3. Predictive Values to Diagnose Sarcopenia at T12 Computed Tomography Level

There are not yet standard cutoff points for the diagnosis of Sc using T12-CT. A ROC analysis was performed, and the cutoff points according to our sample that predict the diagnosis of Sc and low muscle mass can be seen in Table 4. All the cutoff points showed a *p*-value of <0.05, indicating statistical significance (Figure 2).

### 3.4. Correlation Analysis between Muscle Measures: CT, BIVA, NU and Functional Test (HGS)

Table 5 presents the muscle correlation coefficients between CT measurements and BIVA, NU and HGS. SMA_T12CT was highly correlated with BCM (r = 0.785, *p* < 0.001), ASMM (r = 0.761, *p* < 0.001), ASMI (r = 0.786, *p* < 0.001), RF-CSA (r = 0.616, *p* < 0.001), and HGS (r = 0.465, *p* < 0.001). Similarly, SMI_T12CT showed strong correlations with BCM (r = 0.681, *p* < 0.001), ASMM (r = 0.589, *p* < 0.001), ASMI (r = 0.775, *p* < 0.001), RF-CSA (r = 0.599, *p* < 0.001), and HGS (r = 0.350, *p* < 0.001), highlighting its significance in assessing muscle mass and function. The reliability of the scale was assessed using Cronbach’s alpha, which yielded a value of 0.735 (Figure 3A). This indicates acceptable internal consistency for the set of items measured by the scale. 

Additionally, Figure 3B presents the fat mass correlation into the relationships between T12-CT scan parameters (VAT percentage and SAT percentage) with parameters from BIVA (FMI) and NU (T-SAT, L-SAT). T12-CT scan parameters showed that SAT_area_T12 had strong positive correlations with BMI (r = 0.696, *p* < 0.001), FMI (r = 0.778, *p* < 0.001) and L-SAT (r = 0.611, *p* < 0.001), and a strong negative correlation with SAT_HU_T12 (r = −0.682, *p* < 0.001).

### 3.5. Kaplan–Meier Survival Curve in Idiopathic Pulmonary Fibrosis Patients Categorized by Muscle Mass Index

Our findings indicated that patients with low skeletal muscle index (SMI) measured by CT exhibited significantly reduced median survival and higher risk of mortality compared to those with normal muscle mass (SMI cut off ≤ 28.8 cm/m^2^). Specifically, the 12-month survival rate was 85% in low muscle mass patients versus 100% in non-low muscle mass patients (Figure 4). Although the hazard ratio of 6.20 did not reach statistical significance (*p* = 0.079), the trend suggests a considerably increased risk of adverse outcomes associated with low muscle mass. However, the log-rank value of the survival curve does show a moderate significance.

## 4. Discussion

Our results indicate a high correlation between T12-CT derived muscle measurements and other MFA techniques, such as bioelectrical impedance analysis (BIVA), nutritional ultrasound (NU), hand grip strength (HGS), and the timed up and go (TUG) test. The results highlight that low muscle mass, as assessed by T12-CT, is associated with greater disease severity and increased mortality risk in IPF patients.

Malnutrition in IPF patients is frequent [4,5,6]. It is associated with higher mortality, increased hospitalizations, and lower quality of life [5,6,19,20], constituting a prognostic indicator of antifibrotic therapy discontinuation and mortality, independently of age, sex, forced vital capacity, or gender-age-physiology index [20]. Yuji Iwanami et al. demonstrated a moderate risk of malnutrition, measured by the CONUT (Controlling Nutritional Status) Score, in 40% of a sample of 170 patients with similar characteristics to ours, which was associated with a significantly higher risk of all-cause mortality [4]. In our series, we observed that 57.3% of patients were diagnosed with malnutrition, according to the GLIM criteria.

However, other nutritional diagnoses, such as Sc, which involve changes in BC could not only be part of the diagnosis of malnutrition but also an aggravating factor [2,8,22,23,24,25,26]. By diagnosing these conditions, we could offer greater nutritional and personalized management.

Our findings indicate that Sc is associated with older age and lower muscle mass and function, highlighting the importance of nutritional and functional assessments in this population. These findings are consistent with current evidence [11,12,15,16,17,18].

Although Sc defined by the EWGSOP2 criteria [12,13] is widely used, only one value for each criterion, without sex differentiation, was included. We can underscore the importance of sex-specific cut-off points in the assessment of Sc, highlighting significant differences in muscle strength and mass between men and women.

Morphofunctional techniques, such as NU and BIVA, have already been shown to be good prognostic markers of 12-month mortality in IPF patients [28]. Nonetheless, in IPF patients specifically, we believe that thoracic CT is a potential and powerful tool to detect BC impairments, as it is usually performed in routine practice. In fact, the T12 vertebra level was chosen as it is always included in CT imaging, in contrast to L1, which sometimes is not included, as reported by some studies [51,52]. This was also the case in most of the CT scans in our series. Even if CT-L3 is considered the best single-slice CT for BC [29,30], studies that compare both L3-CT and T12-CT showed high correlation in a sample of 239 patients who underwent whole-body PET–CT scans [30], in healthy people [35], geriatric patients, or patients with hematologic cancers [14], pediatric solid tumors [33], SARS-CoV2 infection [36] or those undergoing valvular surgery [37]. In some studies, not only the skeletal muscle mass but also the subcutaneous fat areas were measured [14,36].

The ESMA assessment from T12-CT scans has also been used to evaluate Sc and cachexia in COPD patients [23,24], showing a significant correlation with physiological parameters, symptoms, and disease prognosis. They also confirm ESMA assessed by T12-CT as an independent prognostic factor for COPD patients [23]. Other studies investigate the skeletal muscle area index (SMI) at T12-CT (SMI_T12CT). Cho et al. reported that patients with a smaller value of SMI_T12CT were associated with inferior survival after lung transplantation [51]. A good correlation has also been demonstrated between muscle parameters (SMA and SMI) at T12-CT and L3-CT [14,35].

Regarding IPF, Suzuki et al. found that smaller ESMA is associated with a poor prognosis [24], and Çinkooğlu et al. highlighted that ESMA at T12-CT predicts both short- and long-term survival [11]. Additionally, Jalaber C et al. proposed SMI measured at the L1 level as a trusted tool to exclude malnutrition in IPF patients [53].

Correlations have been observed between RF-CSA via NU and Pha via BIVA in oncology patients [54,55], post-critical COVID-19 patients [56] and IPF patients [28]. Nevertheless, to the best of our knowledge, this is the first study with an MFA including BIVA, NU, T12-CT and HGS in patients with IPF. Results presented indicate strong correlations between CT-derived muscle measurements and other morphofunctional parameters, highlighting the consistency and reliability of these methods in assessing muscle mass and function. Additionally, the relationships, shown in Figure 1, between higher adiposity and lower tissue density are particularly notable, reflecting the complex nature of BC and its assessment. Overall, these correlations highlight the interconnectedness of classic BC measures, BIVA, NU, and CT scan parameters, emphasizing the value of comprehensive BC analysis in clinical settings.

Our findings highlight distinct differences in muscle and adipose tissue distribution and density between sarcopenic and non-sarcopenic individuals, emphasizing the importance of comprehensive BC analysis in this population. Future studies should focus on adipose tissue and density distribution, as myoesteatosis, characterized by the accumulation of fat within muscle tissue, has been linked with worsening of lung function loss in IPF [57] and it may precede Sc, constitute an exacerbated condition or even be present independently of Sc.

Unfortunately, reference values for defining Sc on T12-CT are not yet available [37]. The cut-off value of SMI_T12CT in our sample was 24.5 cm^2^/m^2^ for Sc and 28.8 cm^2^/m^2^ for low muscle mass. As explained in Table 4, our results suggest that while the SMA_T12CT offers high specificity, its sensitivity is relatively low. Conversely, the SMI_T12CT provides a balanced sensitivity and specificity, making it a valuable parameter for diagnosing Sc. Moreover, the results observed in Figure 2 highlight the significance of low muscle SMI_T12CT cut-off of 27.2 cm^2^/m^2^ as a predictor of survival. Higher values of SMI_T12CT are associated with reduced mortality risk and longer survival times. The analysis demonstrates that using a higher cut-off value for SMI_T12CT improves its sensitivity in predicting mortality, emphasizing its importance in clinical assessments. A similar significant low SMI_T12CT cut-off value was reported by Young Hoo Cho et al. They selected from 45 post-lung transplant patients a cut-off SMI_T12CT value of 28.07 cm^2^/m^2^ for diagnosing Sc [51], demonstrating that those with lower values had inferior survival after lung transplantation in both univariate and multivariate analyses [51], and shortened ventilator support and ICU stay periods in those with non-low SMI_T12CT.

In total, 30% of our sample died within the 12-month follow-up period, most (95%) due to a terminal phase of the disease. It is time to focus on new potential survival predictors, such as those mentioned in this article, to confirm their role in improving disease management, aiming for a better quality of life and a longer survival rate for these patients.

The main strength of this study is its comprehensive approach to assessing body composition in patients with IPF through multiple MFA techniques. This multifaceted method ensures a thorough evaluation of muscle mass and function, providing robust data that underscores the interconnectedness of these measures. The study’s use of T12-CT scans, commonly included in routine practice, adds practical relevance, particularly for detecting BC impairments when L1 vertebrae are not captured. Furthermore, the incorporation of a diverse range of assessment tools specifically for IPF patients sets a precedent for future research and clinical practice.

To conclude, the study has several limitations. First, the sample size is relatively modest, which may affect the generalizability of the findings. Specifically, IPF patients without CT scans performed during the study period were excluded, potentially leading to selection bias. Additionally, the study cohort predominantly consists of male patients, reflecting the male predominance of IPF, but this may limit the applicability of the results to female patients. Another limitation is the absence of established reference values for defining Sc using CT, which challenges the interpretation and comparison of the findings. Lastly, while FocusedON^®^ shows promise, it is still a developing methodology, and further studies are necessary to validate and refine its use in clinical practice [49]. Nonetheless, there is also the possibility of confounding factors that we were unable to address in these analyses.

While our study highlights the importance of comprehensive BC analysis in IPF patients, this is a pilot study, and further research with larger, more diverse populations and well-designed prospective studies are needed to confirm these relationships and enhance disease management strategies. The promising results of FocusedON^®^ and other morphofunctional techniques warrant continued investigation to refine these methodologies and improve clinical outcomes for IPF patients.

## 5. Conclusions

This study underscores the importance of a comprehensive approach to assessing BC in patients with IPF. By integrating multiple MFA techniques (BIVA, NU, HGS and TUG) and computed tomography (T12-CT), we have provided robust data highlighting the interconnectedness of these measures.

The use of T12-CT scans is a reliable technique for measuring low muscle mass in patients with IPF. This method is particularly relevant for detecting BC impairments when the L3 vertebrae are not captured, demonstrating high practical applicability and reliability in clinical settings. An SMI value of <28.8 is a good predictor of low lean mass and 12-month mortality in IPF patients.

## Figures and Tables

**Figure 1 nutrients-16-02885-f001:**
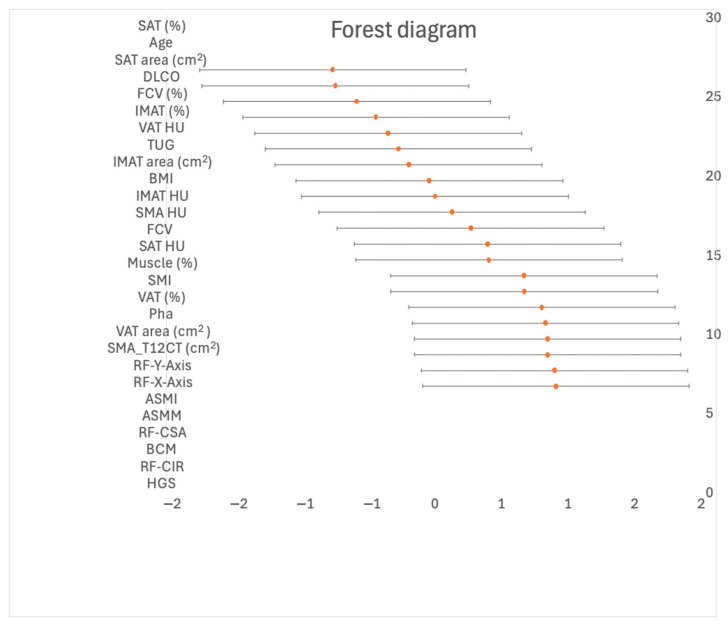
Forest diagram of the sample effect. Abbreviations: BMI: body mass index, DLCO: pulmonary carbon monoxide diffusing capacity, FCV: forced vital capacity, HGS: handgrip strength, TUG: get up and go test, Pha: phase angel, BCM: body mass cell, ASMM: appendicular skeletal muscle mass, ASMI: appendicular skeletal muscle mass index, RF-CSA: rectus femoris cross-sectional area, RF-*Y*-axis: rectus femoris *Y*-axis, RF-*X*-axis: rectus femoris *X*-axis, RF-CIR: circumference of the quadriceps rectus femoris, HU: Hounsfield Units; SMA_T12: Skeletal Muscle Area in T12-CT, SMI_T12CT: Skeletal Muscle Index in T12-CT, IMAT: Intra-Muscular Adipose Tissue; VAT: visceral adipose tissue; SAT: subcutaneous adipose.

**Figure 2 nutrients-16-02885-f002:**
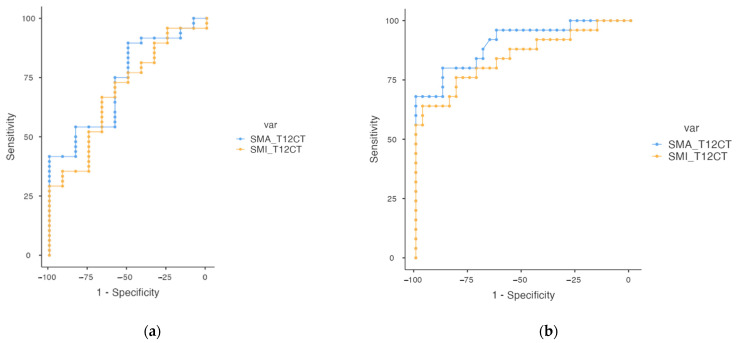
ROC curve cut-off point at T12 in IPF patients. (**a**) Sarcopenia cut-off SMA and SMI; (**b**) Low muscle mass cut-off SMA and SMI. Abbreviations: SMA_T12CT: Skeletal Muscle Area at T12 level, SMI_T12CT: Skeletal Muscle Index at T12 level.

**Figure 3 nutrients-16-02885-f003:**
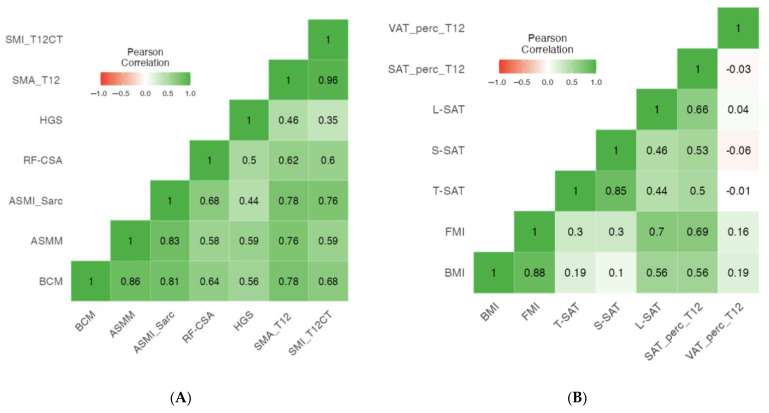
Heatmap of muscle (**A**) and adipose (**B**) tissue parameters with different morphofunctional techniques. Abbreviations: BCM: body mass cell, ASMM: appendicular skeletal muscle mass, ASMI: appendicular skeletal muscle mass index, RF-CSA: rectus femoris cross-sectional area, HGS: handgrip strength, SMA_T12: Skeletal Muscle Area at T12-CT, SMI_T12CT: skeletal mass index at T12-CT. VAT_perc_T12: percentage of visceral adipose tissue at T12 computed tomography level, SAT_perc_T12: percentage of subcutaneous adipose tissue at T12 computed tomography level, L-SAT: leg subcutaneous adipose tissue, S-SAT: superficial abdominal adipose tissue, T-SAT: total abdominal adipose tissue, FMI: fat-free mass index, BMI: body mass index.

**Figure 4 nutrients-16-02885-f004:**
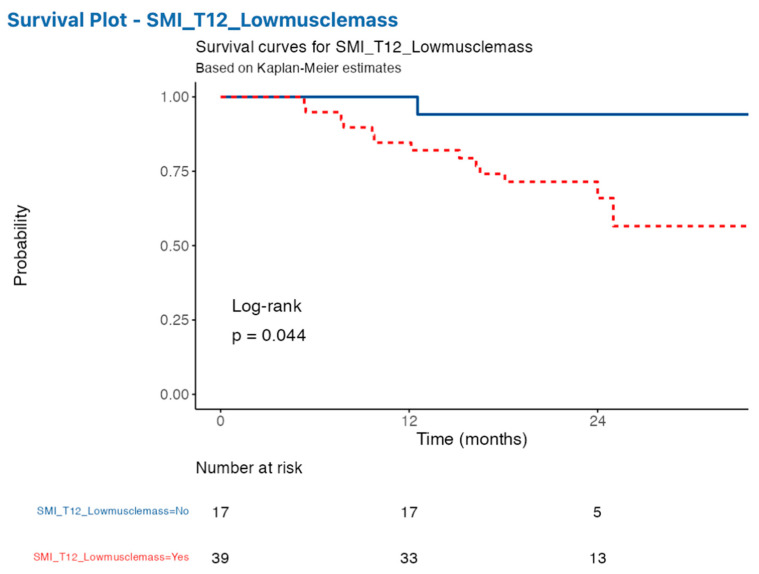
Kaplan–Meier survival curve of IPF patients with low or normal SMI. IPF patients with low SMI (red line) died more frequently than those with non-low SMI (blue line) measured by CT over a 12-month follow-up period. Abbreviations: SMI: skeletal muscle mass. IPF: idiopathic pulmonary fibrosis.

**Table 1 nutrients-16-02885-t001:** Baseline characteristics of our sample population.

	All N = 60	Non-Sarcopenic *n* = 48	Sarcopenic *n* = 12	Effect Size (Hedge’s)	95% IC (Lower)	95% IC (Upper)	*p* Value
Demographic and anthropometric variables							
Age (years)	70.9 ± 7.8	69.7 ± 7.4	75.3 (8.1)	−0.751	−13.951	−0.0978	0.024
Gender (male)	52.0 (85.2%)	46.0 (76.7%)	5.0 (8.3%)				<0.001
BMI (kg/m^2^)	27.7 ± 3.7	27.8 ± 3.7	27.8 (3.8)	0.0153	−0.6141	0.6449	0.969
Respiratory variables							
DLCO (%)	51.2 ± 17.7	48.3 ± 16.0	62.2 (19.9)	−0.57	−12.511	0.1181	0.021
FCV (L)	2638.0 (772.0)	2708.0 ± 742.0	2322 (840.0)	0.411	−0.2332	10.562	0.126
FCV (%)	65.1 ± 15.4	63.0 ± 14.9	68.8 (17.1)	−0.428	−10.705	0.2164	0.315
Functional measurement							
HGS (kg)	33.4 ± 10.2	36.7 ± 8.7	20.8 (4.4)	1.849	10.967	25.834	<0.001
TUG	8.31 ± 5.94	8.1 ± 6.2	9.1 (5.1)	−0,181	−0.8353	0.4753	0.632
BIVA							
Pha	4.8 ± 0.7	4.9 ± 0.7	4.5 ± 0.7	0,847	0.1882	14.973	0.130
BCM	26.3 ± 5.18	27.3 ± 5.1	22.3 ± 3.2	1.267	0.5784	19.466	<0.05
ASMM (kg)	20.6 ± 3.2	21.2 ± 3.2	17.8 ± 1.1	1.210	0.5245	18.821	0.014
ASMI (kg/m^2^)	7.2 ± 0.7	7.3 ± 0.8	6.6 ± 0.4	1.158	0.4778	18.268	0.139
NU							
RF-CSA (cm^2^)	3.3 ± 1.0	3.6 ± 1.1	2.5 ± 0.5	1.240	0.5502	19.181	<0.001
RF-*Y*-axis (cm)	1.1 ± 0.2	1.2 ± 0.3	1.0 ± 0.2	1.428	0.7215	21.225	0.022
RF-*X*-axis (cm)	3.4 ± 0.4	3.5 ± 0.4	3.1 ± 0.5	0.926	0.2603	15.815	0.002
RF-CIR (cm)	8.16 ± 1.0	8.4 ± 0.9	7.1 ± 0.9	1.428	0.7215	21.225	<0.001
Malnutrition after follow-up (GLIM) (%)							<0.05
Non malnutrition	42.6%	31.7%	11.7%				
Moderate malnutrition	31.1%	30.0%	0.0%				
Severe malnutrition	26.2%	18.3%	8.3%				
Mortality (%)							<0.05
No	70%	50.0%	20.0%				
Yes	30%	30.0%	0.0%				

Data included are expressed as means ± standard deviations, percentages or absolute numbers. Groups were divided according to diagnosis of sarcopenia. Abbreviations: IC: interval confidence; BMI: body mass index, DLCO: pulmonary carbon monoxide diffusing capacity, FCV: forced vital capacity, HGS: handgrip strength, TUG: get up and go test, BIVA: bioelectrical impedance vectorial analysis, Pha: phase angel, BCM: body mass cell, ASMM: appendicular skeletal muscle mass, ASMI: appendicular skeletal muscle mass index, NU: nutritional ultrasound, RF-CSA: rectus femoris cross-sectional area, RF-*Y*-axis: rectus femoris *Y*-axis, RF-*X*-axis: rectus femoris *X*-axis, RF-CIR: circumference of the quadriceps rectus femoris.

**Table 2 nutrients-16-02885-t002:** Sarcopenia according to EWGSOP2 criteria.

		N = 60	*p*
Handgrip strength (kg)			
Total	Mean ± SD	33.4 ± 10.2	<0.001
Men	Mean ± SD	35.3 ± 8.9	
Women	Mean ± SD	22.7 ± 10.6	
Low handgrip strength	Mean (%)	19 (31.7%)	
ASMM (kg)			
Total	Mean ± SD	20.6 ± 3.2	0.014
Men	Mean ± SD	21.0 ± 3.1	
Women	Mean ± SD	18.2 ± 2.6	
Low ASMM	Mean (%)	30 (49.2%)	0.009
ASMI (kg/talla)			
Total	Mean ± SD	7.2 ± 0.8	0.139
Men	Mean ± SD	7.3 ± 0.8	
Women	Mean ± SD	6.8 ± 0.4	
Low ASMI	Mean (%)	28 (45.9%)	0.182
Total low muscle massbb (low ASMI or ASMM)	Mean (%)	32 (56.1%)	
Sarcopenia(Low HGS and Los muscle mass)	Mean (%)	12 (20%)	

Abbreviations: ASMM: appendicular skeletal muscle mass, ASMI: appendicular skeletal muscle mass index.

**Table 3 nutrients-16-02885-t003:** Differences in body composition parameters by T12-CT according to sarcopenia.

T12-CT Parameters	All (N = 60)	Non Sarcopenic (*n* = 48)	Sarcopenic (*n* = 12)	Effect Size (Hedge’s)	95% IC (Lower)	95% IC (Upper)	*p*
SMA_T12CT (cm^2^)	75 ± 21.8	78.8 ± 22.3	60.6 ± 12.9	0.8604	0.2084	15.040	0.009
Muscle (%)	9.5 ± 2.1	9.9 ± 2.2	8.4 ± 1.7	0.6837	0.041	13.204	0.036
Muscle (HU)	39.0 ± 7.2	39.5 ± 7.6	37.4 ± 5.6	0.2841	−0.3445	0.9103	0.376
SMI_T12CT (cm^2^/m^2^)	26.2 ± 6.9	27.2 ± 7.1	22.6 ± 4.8	0.6862	0.0433	13.217	0.035
IMAT area (cm^2^)	14.9 ± 6.8	14.8 ± 6.9	15.0 ± 6.7	−0.0264	−0.6501	0.5983	0.934
IMAT (%)	1.87 ± 0.7	1.8 ± 0.7	2.1 ± 0.8	−0.3363	−0.9622	0.2933	0.296
IMAT (HU)	−63.9 ± 5.4	−63.7 ± 5.5	−64.5 ± 5.4	0.1443	−0.4822	0.7684	0.652
VAT area (cm^2^)	177.0 ± 81.6	191.5 ± 84.8	123.5 ± 36.3	0.8604	0.2084	15.040	0.009
VAT (%)	22.2 ± 8.4	23.6 ± 8.7	16.9 ± 4.2	0.8176	0.1681	14.592	0.013
VAT (HU)	−97.7 ± 6.3	−98.0 ± 6.0	−96.3 ± 7.7	−0.2584	−0.8838	0.3694	0.420
SAT area (cm^2^)	119 ± 56.6	111.6 ± 48.3	152.4 ± 77.3	−0.2584	−0.8838	0.3694	0.025
SAT (%)	15 ± 6.5	13.6 ± 4.8	20.8 ± 9.6	−11.875	−18.491	−0.5142	<0.001
SAT HU	−98.5 ± 9.7	−97.6 ± 9.2	−101.7 ± 11.5	0.4207	−0.2115	10.497	0.192

Abbreviations: HU: Hounsfield Units; SMA_T12: Skeletal Muscle Area in T12-CT, SMI_T12CT: Skeletal Muscle Index in T12-CT, IMAT: Intra-Muscular Adipose Tissue; VAT: visceral adipose tissue; SAT: subcutaneous adipose.

**Table 4 nutrients-16-02885-t004:** Predictive value to diagnose sarcopenia and low muscle mass in T12-CT.

	Variables	Cut-Off	AUC	Sensitivity	Specificity	Youden’s Index	*p*
Sarcopenia	SMA_T12CT	77.4	0.734	41.7%	100%	0.417	<0.05
	SMI_T12CT	24.5	0.689	66.7%	66.7%	0.333	<0.05
Low muscle mass	SMA_T12CT	80.5	0.904	68.0%	100.0%	0.680	<0.05
	SMI_T12CT	28.8	0.848	64.0%	96.8%	0.609	<0.05

Abbreviations: SMA_T12CT: Skeletal Muscle Area in T12 computed tomography level, SMI_T12CT: Skeletal Muscle Index in T12 computed tomography level, AUC: area under the curve.

**Table 5 nutrients-16-02885-t005:** Correlation CT with other morphofunctional tests (BIVA, NU and HGS).

	BCM	ASMM	ASMI	RF-CSA	HGS	Muscle_Area_T12
BCM	—					
ASMM	0.864 ***	—				
ASMI	0.810 ***	0.825 ***	—			
RF-CSA	0.637 ***	0.575 ***	0.679 ***	—		
HGS	0.560 ***	0.592 ***	0.441 ***	0.497 ***	—	
SMA_T12CT	0.785 ***	0.761 ***	0.786 ***	0.616 ***	0.465 ***	—
SMA_perc_T12	0.591 ***	0.478 ***	0.562 ***	0.528 ***	0.373 **	0.831 ***
SMI_T12CT	0.681 ***	0.589 ***	0.775 ***	0.599 ***	0.350 **	0.956 ***

Abbreviations: ** *p* < 0.01, *** *p* < 0.001, BCM: body mass cell, ASMM: appendicular skeletal muscle mass, ASMI: appendicular skeletal muscle mass index, RF-CSA: rectus femoris cross-sectional area, HGS: handgrip strength, SMA_T12: Skeletal Muscle Area in T12-CT, SMA_perc_T12: percentage of skeletal muscle area at T12-CT, SMI_T12CT: skeletal mass index at T12-CT.

## Data Availability

The original contributions presented in the study are included in the article/Appendix A, further inquiries can be directed to the corresponding author.

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
