# Peer review of "IA-Body Composition CT at T12 in Idiopathic Pulmonary Fibrosis: Diagnosing Sarcopenia and Correlating with Other Morphofunctional Assessment Techniques"

_nutrients, 2024, doi:10.3390/nu16172885_

Round 1
Reviewer 1 Report
Comments and Suggestions for Authors
The authors evaluated the associations of skeletal mass measures from CT with measures of sarcopenia in patients with IPF to explore the prognostic value of these novel measures in IPF patients. The topic is of great clinical importance. Thank you for the opportunity to review this submission. Suggestions:
1] Abstract: in the first few lines (before methods) please state a clear and explicit hypothesis or hypotheses being evaluated or state a clear aim for this study.
2] Introduction is too long with a lot of information. Please shorten or remove introductory sentences for diseases other than IPF; especially because there are several relevant papers for IPF patients which is the focus of this study.
3] Please mention and cite following relevant papers on this topic:
(1) Jalaber C, Lemerre-Poincloux J, Jouneau S, Rousseau C, Dolou B, Rouag E, Lescoat A, Luque-Paz D, Lucas C, Vernhet L, Thibault R, Lederlin M. Usefulness of Body Composition CT Analysis in Patients with Idiopathic Pulmonary Fibrosis: A Pilot Study. Acad Radiol. 2022 Feb;29 Suppl 2:S191-S201. doi: 10.1016/j.acra.2021.07.020. Epub 2021 Aug 18. PMID: 34417107.
(2) Nakano A, Ohkubo H, Taniguchi H, Kondoh Y, Matsuda T, Yagi M, Furukawa T, Kanemitsu Y, Niimi A. Early decrease in erector spinae muscle area and future risk of mortality in idiopathic pulmonary fibrosis. Sci Rep. 2020 Feb 11;10(1):2312. doi: 10.1038/s41598-020-59100-5. PMID: 32047177; PMCID: PMC7012911.
(3) Awano N, Inomata M, Kuse N, Tone M, Yoshimura H, Jo T, Takada K, Sugimoto C, Tanaka T, Sumikawa H, Suzuki Y, Fujisawa T, Suda T, Izumo T. Quantitative computed tomography measures of skeletal muscle mass in patients with idiopathic pulmonary fibrosis according to a multidisciplinary discussion diagnosis: A retrospective nationwide study in Japan. Respir Investig. 2020 Mar;58(2):91-101. doi: 10.1016/j.resinv.2019.11.002. Epub 2019 Dec 24. PMID: 31882370.
4] Please state an explanation for choosing CT at T12 level and not any other level.
5] Please provide references for section 2.2.5
6] One conclusion is (line 52-53): "An ASMI value of < 28.8 is a good predictor of low lean mass and 12-month mortality in IPF patients." For supporting such a statement, please consider providing predictive statistic such as the Youden's index.
7] In the limitations para, please also discuss potential for confounding which was not addressed in these analyses. Please mention that this is an exploratory/pilot study and that these finding need to be confirmed in future well-designed larger prospective studies.
Thank you for your contribution.
Author Response
Estimado revisor,
Nos gustaría comenzar expresando nuestro más sincero agradecimiento por su minuciosa revisión de nuestro manuscrito. A continuación encontrará nuestras respuestas detalladas y las correcciones correspondientes resaltadas en los archivos reenviados.
Comentario 1: Resumen: en las primeras líneas (antes de los métodos) indique una hipótesis o hipótesis claras y explícitas que se están evaluando o indique un objetivo claro para este estudio.
Respuesta 1: Gracias por señalar esto. Hemos revisado el resumen y hemos añadido la siguiente frase justo antes de la sección de métodos: “Nuestro objetivo era explorar la correlación de BIVA, NU y parámetros funcionales con el CC en tomografías computarizadas de nivel T12 en pacientes con FPI, pero también su relación con el grado de SC, la desnutrición y la mortalidad” (líneas 39-41).
Comentario 2: La introducción es demasiado larga y contiene demasiada información. Por favor, acorte o elimine las oraciones introductorias para enfermedades distintas de la FPI, especialmente porque hay varios artículos relevantes para pacientes con FPI, que es el foco de este estudio.
Respuesta 2: Estamos totalmente de acuerdo con tu observación, sobre todo teniendo en cuenta la necesidad de incluir los artículos mencionados en tu comentario siguiente. Por ello, hemos acortado la introducción eliminando algunas frases y condensando las que tratan sobre otras patologías.
Comentario 3: Por favor, mencione y cite los siguientes artículos relevantes sobre este tema:
(1) Jalaber C, Lemerre-Poincloux J, Jouneau S, Rousseau C, Dolou B, Rouag E, Lescoat A, Luque-Paz D, Lucas C, Vernhet L, Thibault R, Lederlin M. Utilidad del análisis de la composición corporal por TC en pacientes con fibrosis pulmonar idiopática: un estudio piloto. Acad Radiol. 2022 feb;29 Suppl 2:S191-S201. doi: 10.1016/j.acra.2021.07.020. Publicación electrónica 18 de agosto de 2021. PMID: 34417107.
(2) Nakano A, Ohkubo H, Taniguchi H, Kondoh Y, Matsuda T, Yagi M, Furukawa T, Kanemitsu Y, Niimi A. Disminución temprana del área del músculo erector de la columna y riesgo futuro de mortalidad en la fibrosis pulmonar idiopática. Sci Rep. 11 de febrero de 2020;10(1):2312. doi: 10.1038/s41598-020-59100-5. PMID: 32047177; PMCID: PMC7012911.
(3) Awano N, Inomata M, Kuse N, Tone M, Yoshimura H, Jo T, Takada K, Sugimoto C, Tanaka T, Sumikawa H, Suzuki Y, Fujisawa T, Suda T, Izumo T. Medidas cuantitativas de la masa muscular esquelética mediante tomografía computarizada en pacientes con fibrosis pulmonar idiopática según un diagnóstico de discusión multidisciplinaria: un estudio retrospectivo a nivel nacional en Japón. Respir Investig. 2020 Mar;58(2):91-101. doi: 10.1016/j.resinv.2019.11.002. Publicación electrónica 24 de diciembre de 2019. PMID: 31882370.
Respuesta 3: Agradecemos sinceramente su recomendación de incluir estos estudios pertinentes. Hemos incorporado un nuevo párrafo al final de la introducción, justo antes de nuestros objetivos, que dice: “En pacientes con FPI, se encontró que una ESMCSA baja en T12-CT era un predictor significativo de mortalidad por todas las causas (38), lo que resalta su importancia como indicador pronóstico clave, superando a otros como el IMC y la CVF (39)”. Este párrafo hace referencia a los estudios de Nakano et al. y Awano et al. (líneas 116-118).
Respecto al estudio de Jalaber et al., hemos decidido mencionarlo en la sección de discusión, donde hablamos del índice de músculo esquelético, con la esperanza de que esta ubicación cuente con su aprobación (líneas 426-427).
Comentario 4: Indique una explicación por qué eligió CT en el nivel T12 y no en cualquier otro nivel.
Respuesta 4: Estamos de acuerdo con su sugerencia. Si bien abordamos este tema en la sección de discusión (líneas 407-409), coincidimos en que debería aclararse antes en el manuscrito. Para enfatizar este punto, hemos agregado lo siguiente:
- En la introducción (líneas 111-113): “Sin embargo, la TC L3 está disponible en la TC abdominal pero no en la TC torácica (que se utiliza a menudo para el seguimiento de pacientes con FPI), por lo que algunos estudios han comparado la TC L3 y las TC a nivel T12 (TC T12), ya que es la vértebra más cercana a L3 la que siempre se captura en la TC torácica, buscando la mejor vértebra torácica para el CC (14,30,33,35–37)”.
- En la sección 2.2.5. (líneas 213-218): “Se eligió T12-CT por ser la vértebra más cercana a L3 que se captura de manera consistente en las tomografías computarizadas torácicas, a diferencia de L1 o L2. Además, a este nivel, no hay grandes vasos, como la aorta torácica, que incluyen músculo en sus estructuras, lo que podría interferir con nuestro software, a diferencia del nivel T4, que también se ha explorado en algunos estudios”.
Comentario 5: Proporcione referencias para la sección 2.2.5
Respuesta 5 Entiendo la petición de más referencias para justificar el uso de este software, pero es un software nuevo en el mercado, validado por el grupo de investigación del hospital Vall d'Hebron, he podido añadir otra publicación del grupo en pacientes con cáncer de colon utilizando este software pero en el corte L3, esperamos que sea suficiente:
- Soria-Utrilla V, Sánchez-Torralvo FJ, Palmas-Candia FX, Fernández-Jiménez R, Mucarzel-Suarez-Arana F, Guirado-Peláez P, et al. Evaluación de la composición corporal asistida por IA mediante imágenes de TC en pacientes con cáncer colorrectal: capacidad predictiva para el diagnóstico de sarcopenia y desnutrición. Nutrientes. enero de 2024;16(12):1869.
Comentario 6: Una conclusión es (líneas 52-53): "Un valor ASMI de < 28,8 es un buen predictor de baja masa magra y mortalidad a los 12 meses en pacientes con FPI". Para respaldar tal afirmación, considere proporcionar estadísticas predictivas como el índice de Youden.
Respuesta 6: Creo que es un buen añadido para reforzar este dato, se ha añadido en la tabla 4 en una columna a la derecha de la especificidad de todos los puntos de corte calculados. También hemos cambiado la abreviatura de ASMI por SMI, había un error conceptual. Gracias por vuestra comprensión (Línea 52 y 505).
Comentario 7: En el párrafo sobre limitaciones, analice también la posibilidad de que se produzcan factores de confusión que no se aborden en estos análisis. Mencione que se trata de un estudio exploratorio/piloto y que estos hallazgos deben confirmarse en futuros estudios prospectivos más amplios y bien diseñados.
Respuesta 7 : Estamos de acuerdo con su recomendación y hemos revisado la sección de limitaciones para incluir estos puntos. Hemos hecho hincapié en la naturaleza exploratoria de nuestro estudio y en la necesidad de realizar estudios prospectivos más amplios y bien diseñados en el futuro para confirmar nuestros hallazgos.
Por último, esperamos que nuestra revisión haya mejorado el manuscrito a su entera satisfacción. Hemos adjuntado una versión revisada del artículo, con todos los cambios resaltados para facilitar su revisión.
Gracias una vez más por sus valiosos comentarios.

Reviewer 2 Report
Comments and Suggestions for Authors
Fernández-Jiménez et al present a fine work including 3 different pivotal aspects: a morbidity entity of high interest, a manifestation if not comorbidity, and 3 different diagnostic techniques. This prospective study is prudently designed and performed. The statistics are
Most importantly, due to the very small sample included (sarcopenic patients 12),
a. The survival curve p-value was 0.045, which should be interpreted as "limited significance" rather than "not significant". The correct interpretation is that either the one or more other environmental factors (i.e. room temperature or humidity) or one or more intimate factors affected the result, or intrinsic factor not considered (i.e. genetic factors, clinical or demographic), influenced this outcome.
b. they should convert data into the Hedge's g effect size and 95% CI instead of median or mean and SD and p-values. This is recommended for limiting bias due to small size samples, and assuring the biggest accuracy:
https://stats.stackexchange.com/questions/1850/difference-between-cohens-d-and-hedges-g-for-effect-size-metrics
c. Figures should be added (i.e. ROC curves (the information of the relevant table should be included in the figure legends and/or in text) and the rest in forest plots showing Hedge's g.
Author Response
Dear Reviewer,
We would like to begin by expressing our sincere gratitude for your thorough review of our manuscript. Please find below our detailed responses and the corresponding corrections highlighted in the re-submitted files.
Comment 1: a. The survival curve p-value was 0.045, which should be interpreted as "limited significance" rather than "not significant". The correct interpretation is that either the one or more other environmental factors (i.e. room temperature or humidity) or one or more intimate factors affected the result, or intrinsic factor not considered (i.e. genetic factors, clinical or demographic), influenced this outcome.
Response 1: When we comment in the text (line 371) that there is no significance, we are referring to the significance value of the HR, but if you prefer it we could comment that the survival curve does have a moderate significance.
Comment 2: b. they should convert data into the Hedge's g effect size and 95% CI instead of median or mean and SD and p-values. This is recommended for limiting bias due to small size samples, and assuring the biggest accuracy:
Response 2: Thank you very much for showing me this way of representing samples with fewer patients. We are not sure if we have been able to do it the way you ask us, so we have left it added in the tables (1,3) where the means and medians and the p are and when we confirm that it is correct we will proceed to delete the data that we do not see necessary. As our statistical programme gave us the value of Cohen's d, we have transformed it from a formula that we show you below. And with these values and those of IC we have represented them in the diagram forest that we were asked for.
Formula used:
Where:
d is Cohen's d value.
N=df+2 is the total sample size.
Comment 3: c. Figures should be added (i.e. ROC curves (the information of the relevant table should be included in the figure legends and/or in text) and the rest in forest plots showing Hedge's g.
https://stats.stackexchange.com/questions/1850/difference-between-cohens-d-and-hedges-g-for-effect-size-metrics
Response 3: We think it is interesting to add a visual representation of the data. We have added in the manuscript :
- Figure 1. Forest diagram of the sample effect (line 303).
- Figure 2. ROC curve cut-off point at T12 in IPF patients. (a) Sarcopenia cut-off SMA and SMI; (b) Low mus-cle mass cut-off SMA and SMI (line 322).
We have therefore modified the order of the former figure 1 and figure 2 to the current figure 3 (line 351) and figure 4 (line 373) respectively.
Finally, we hope that our revisions have improved the manuscript to your satisfaction. We have attached a revised version of the article, with all changes highlighted to facilitate your review.
Thank you once again for your valuable feedback.

Round 2
Reviewer 2 Report
Comments and Suggestions for Authors
Approved for publication